# Assessment of Binge Eating Behavior, Body Shape Concerns, and Associated Factors among Female Adolescents of Northern Saudi Arabia: A Cross-Sectional Study

**DOI:** 10.3390/nu16183082

**Published:** 2024-09-13

**Authors:** Ahmed M. Alhuwaydi, Ayidh Muflih Alqahtani, Razan Saud Alsadun, Ohud Saud Alruwaili, Ashokkumar Thirunavukkarasu, Doaa Mazen Abdel-Salam, Yousef Salman A. Alanazi, Ibrahim Ahmed Mahmoud Al-Huwaidi, Rakan Mohammed Ahmed Alhuwaydi

**Affiliations:** 1Department of Internal Medicine, Division of Psychiatry, College of Medicine, Jouf University, Sakaka 72388, Saudi Arabia; 2Department of Conservative Dental Treatment, Dental Specialty Center, Houta Bani Tamim 16511, Saudi Arabia; aalqahtani391@moh.gov.sa; 3Department of General Dental Care, Specialized Dental Center, Turaif 75211, Saudi Arabia; razan.2019qw@gmail.com; 4Department of Public Health and Community Health, Aljouf Health Cluster, Ministry of Health, Sakaka 42421, Saudi Arabia; wad511@hotmail.com; 5Department of Family and Community Medicine, College of Medicine, Jouf University, Sakaka 72388, Saudi Arabia; ashokkumar@ju.edu.sa; 6Department of Public Health and Community Medicine, Faculty of Medicine, Assiut University, Assiut 71515, Egypt; doaa.mazen@aun.edu.eg; 7College of Medicine, Jouf University, Sakaka 72388, Saudi Arabia; yal3nezy@yahoo.com; 8Department of Public Health, Prince Miteb Bin Abdulaziz Hospital, Sakaka 72311, Saudi Arabia; ialhuwaydi@moh.gov.sa; 9Department of Day Surgery, King Abdulaziz Specialist Hospital, Sakaka 72311, Saudi Arabia; ralhuwaydi@moh.gov.sa

**Keywords:** binge eating, body shape, dissatisfaction, Saudi, adolescents, females

## Abstract

Globally, binge eating behavior has emerged as a significant public health concern, especially among female adolescents. Body shape concerns in female adolescents can lead to body dissatisfaction and other mental health issues. Using a cross-sectional study design, we evaluated the frequency of binge eating behavior, body shape concerns, and associated factors among 400 female adolescents. We utilized a pretested Arabic binge eating scale (BES) and a body shape questionnaire—shorter version (BSQ-8C) to collect the required data. We performed Spearman’s correlation analysis to find the strength and direction of the correlation between the BES and BSQ-8C scores. Finally, we applied binomial logistic regression analysis to identify the predictors of body shape concerns. Of the studied participants, 5.5% and 6.2% had medium and severe binge eating behavior. We found a significant positive correlation between the BES and BSQ-8C scores (rho = 0.434, *p* < 0.001). Also, we found that body shape concerns were significantly higher among the monthly family income category of 5000 to 7000 SAR (*p* = 0.005), the severe binge eating categories (*p* = 0.009), and obese adolescents (*p* = 0.001). The present study results can be applied to the development of focused interventions and strategies to address these concerns in this group.

## 1. Introduction

The adolescent period consists of rapid physical, cognitive, and psychosocial growth, which impacts the adolescents’ emotions, cognition, decision-making, and interactions with the people around them [1,2]. The World Health Organization (WHO) defines adolescence as the period spanning from 10 to 19 years of age [3]. During this distinct phase of human growth, it is crucial to establish the foundations for long-term well-being. Furthermore, adolescence is a transitional period that requires mental health support, as more than half of mental health issues develop at these stages and many of them continue throughout adulthood [2,4].

Binge eating behavior is defined by recurrent episodes of binge eating in which a person consumes comparatively large quantities of food in a short period of time and loses control over their eating habits [5]. Globally, binge eating behaviors have been recognized as significant public health issues, particularly among adolescents. Studies conducted across different regions, such as North America, Europe, and Asia, have consistently demonstrated a high prevalence of binge eating behavior and associated psychosocial problems [6,7,8,9]. Specifically, the prevalence of binge eating behavior among females increased from 3.5% in 2000–2006 to 7.8% in 2013–2017, making this condition an alarming issue for public health services and healthcare providers [10]. The prevalence of this disorder has risen with increases in body weight and obesity, which are the common comorbidities leading to the development of several other comorbid problems, including general medical disorders such as diabetes, obesity, hypertension, and chronic pain [11,12].

This behavior has emerged as a significant public health concern in the Kingdom of Saudi Arabia (KSA), especially among female adolescents. Despite extensive research globally, limited studies have been conducted in Middle Eastern countries, including the KSA, where cultural and societal norms significantly influence dietary behaviors [13,14]. Previous studies from the KSA, such as a study conducted in 2018, have reported that the prevalence of binge eating behavior was 25.47% among female adolescents in Arar city in the KSA. That study used the EAT-26 to assess the binge eating behavior of 314 female adolescents. The results of that study also showed that the prevalence of binge eating behavior was relatively high in individuals who were overweight and obese [15]. However, these studies remain fragmented and do not comprehensively understand how these behaviors interact with other sociocultural factors unique to this region. Saudi Arabia has witnessed a surge in obesity rates, with a considerable proportion of the adolescent population classified as overweight or obese [16,17]. The combination of the rising prevalence of binge eating behavior and obesity status has worsened the situation further. 

The existence of binge eating behavior has been linked to a wide range of adverse social and interpersonal outcomes, as well as lower quality of life in both the overweight and obese adolescent populations [18,19]. Ranzenhofer et al. conducted research on a group of overweight teenagers and found that those who engaged in binge eating reported significant limitations in the areas of health, mobility, and self-esteem when compared with their non-binge eating peers, even when accounting for body composition [19]. Furthermore, two studies conducted on general population samples have reproduced comparable findings, demonstrating that symptoms of binge eating or overeating were significantly linked to social dysfunction and family strain in both males and females [18,20]. A study found 18.8% of the university students at Saudi Public University to have binge eating disorders [21]. 

Body shape is a multidimensional construct and is influenced by several factors. Body shape concerns in females are another psychosocial concern among adolescents and can lead to body dissatisfaction and other mental health issues [16,22]. In the era of social media, body shape concerns have risen significantly across countries. According to numerous studies, mental health disorders are most likely to be connected to physical disorders in children and adolescents, including depression, anxiety, oppositional disorders, and different eating disorders [23,24,25]. Binge eating and body shape concerns are believed to arise from the interaction of multiple risk factors. Previous studies have found that sociocultural factors such as education, parents’ income, occupation, and social media influences are associated with these among adolescents [26,27,28]. A recent study conducted by Mallaram GK et al.in India documented a significant relationship between binge eating behavior and body image concerns. Furthermore, they found that the participants’ body mass indexes (BMIs) were significantly associated with these conditions [27].

Although binge eating disorders typically manifest in adolescence, there has not been enough research carried out on these disorders’ diagnosis and management in this age group. The Aljouf region, situated in the northern KSA, presents a unique cultural context characterized by its conservative Islamic values and relatively traditional lifestyle. Continuous monitoring of epidemiological characteristics related to binge eating behavior and body shape concerns is critical to implementing necessary public education measures. However, these authors did not find sufficient evidence that supports implementing health education measures that are tailored to this region. Hence, this study evaluated the frequency of binge eating behavior, body shape concerns, and associated factors among the female adolescent population in Aljouf, the KSA.

## 2. Materials and Methods

### 2.1. Study Description

The present cross-sectional survey was conducted in the Aljouf region of the KSA from May 2023 to January 2024 among the female adolescent population (between 10 and 19 years). We fixed these age criteria according to the World Health Organization (WHO) guidelines [3]. As per the KSA government statistics, there are about 40 thousand female adolescents in this region. We included all female adolescents from the Aljouf region who fell within the age group specified by the WHO. The exclusion criteria comprised adolescents residing outside the Aljouf province and individuals with pre-existing physical or mental disabilities. Furthermore, we excluded those who were unaccompanied by a parent(s) or legal guardian.

### 2.2. Sampling Strategies

The number of participants (*n*) needed for this survey was determined using the WHO sample size calculator. Taking into account the variation in the prevalence of binge eating behaviors across the different surveys, an anticipated proportion of 50% (*p*) was utilized to calculate the necessary sample size. Other criteria used were an 80% power of this study, a 95% confidence interval (CI), and a 5% margin of error. With these values, it was determined that a minimum of 384 female adolescents were required for the current research, which was rounded up to 400. The survey team employed a convenience sampling method to recruit the participants. In this method, the data acquisition team invited the female adolescents from public locations, including shopping complexes, parks, etc. In order to obtain the best possible representative sample, recruitment was performed at different times of the day and on both weekdays and weekends. Furthermore, we restricted data collection to a maximum of 20 participants per day to ensure the data were collected over a period of time.

### 2.3. Data Collection Procedures 

We obtained the institutional review board’s (IRB’s) approval for this research project from Qurrayat Health Affairs, Ministry of Health (Approval no: 2023-68, Dated 30 April 2023). This IRB is registered with the National Committee of Bioethics, the KSA (Reg no: H-13-S-071). This study complied with the Declaration of Helsinki. Initially, the data collectors, who were trained in a standardized way to collect data, briefed the participants and accompanying adults (parents or legal guardians) on the survey’s purpose. Informed consent was obtained from either the parents or legal guardians. The data collection form (Arabic) of the present study consisted of three sections. The first section gathered data related to sociodemographic and health-related characteristics, including age, gender, education phase, media influence on body concerns, measures taken by the adolescent girls for body image maintenance, height, and weight of the participant. Furthermore, the authors measured the participants’ height (cm) and weight (kg) using the standard guidelines [29,30]. BMI status was categorized according to the WHO guidelines, and various authors have used the same in different studies in the KSA [31,32,33]. The second division inquired about adolescents’ eating habits using the binge eating scale (BES). The BES encompasses 16 questions, each with 3–4 separate responses (items 6 and 16 consisted of three responses, while the other items had four responses), and each is assigned a numerical value (ranging from 0 to 3), which reflects a range of severity from no problems (scored as 0) to severe problems (scored as 3). Therefore, the overall score from the 16 questions ranged from 0 to 46. A score of 27 or higher is conventionally used as a threshold for identifying severe binge eating, while a score of 17 is used to identify mild or no binge eating. This 16-item BES is valid, reliable, and used in various settings across countries [34,35,36]. 

The final section utilized the body shape questionnaire—shorter version (BSQ-8C) to evaluate the participants’ body image concerns. The BSQ-8C is a shortened version of the original instrument (BSQ-34) that maintains adequate psychometric properties. The BSQ-34 was created by Cooper et al. to measure concerns about body shape [37]. The BSQ-8C is a standard and validated tool widely used in research settings to determine body image dissatisfaction and concerns. The instruments consist of 8 items, and the participant responses in each item range from not at all (0 points) to all the time (6). These eight items cover various aspects of body dissatisfaction, including feelings about specific body parts, concern about weight gain, and overall dissatisfaction with body shape. We computed the overall scores of the BSQ-8C and categorized them into no (<19), mild (19 to 25), moderate (26 to 33), and severe (>33) body concerns [25,38,39]. The BES and BSQ-8C were pretested during the pilot study on 30 eligible female adolescents. The Cronbach alphas, measures of internal consistency, for the BES and BSQ-8C were 0.83 and 0.77, respectively.

### 2.4. Data Analysis

The collected data were coded and analyzed using Statistical Package Version 24 (SPSS V.24). The participants’ descriptive data were depicted as frequency and percentages, and the continuous data are presented as means and standard deviation (SD). The Shapiro–Wilk test revealed that the BES and BSQ-8C did not meet the normality assumption of data. Therefore, we analyzed the strength and correlation between these two sections using Spearman’s correlation test (non-parametric). Finally, the associated factors for the body shape concerns were evaluated using a binomial logistic regression analysis. This analysis determined the adjusted odds ratio (AOR) and two-tailed significant value (*p*-value). An independent variable with a *p*-value of less than 0.05 was considered to have a significant association with the outcome variable.

## 3. Results

During the data collection period, we contacted 451 eligible adolescent girls to obtain the minimum required sample size (response rate: 88.7%). Of the 400 studied participants, 51.0% were less than 17 years old, with a mean ± SD of 15.47 ± 2.83, and 56.3% were studying at the high school level or below. Regarding family background characteristics, more than half (56.0%) of the fathers worked in the government sector, and 65% of the fathers and 60.7% of the mothers studied to the college level and above. We found that more than one-third (42.0%) of the adolescent girls’ self-perceptions of their body images were influenced by the media, including social media. Regarding obesity status, more than two-thirds (76.3%) of the studied population were not obese or overweight (Table 1).

Regarding the measures by the adolescent girls of their body image maintenance, 22.5% were on special exercise to reduce their weight and 11.0% were on special diets to reduce their weight (Figure 1). 

Of the 400 participants, 78.2% had no problem controlling the impulse to eat, and 85.3% of them were able to stop eating when they wanted. We also found that nearly half (52.7%) of the participants did not have the habit of eating when they were bored. The detailed responses of the participants in the BES are presented in Appendix A.

Appendix A depicts the participants’ responses in the BSQ-8C. Of the participants, nearly one-fourth (25.5%) had self-awareness of their looks all the time while they were in the company of others, 12.3% responded that feeling full after a large meal made them feel obese, and 8% had concerns about becoming fat. The lowest levels of concern were observed in consideration of surgical procedures such as gastrectomies or removal of body fat (1%).

Regarding binge eating behavior, we found that 6.2% and 5.5% of the participants had high and medium binge eating problems. The mean BES score was 13.27, with a standard deviation (SD) of 4.13 (Table 2). 

We found that 8.5%, 7.5%, and 7.8% of the female adolescents in the present study had mild, moderate, and severe body shape concerns, respectively. The mean body shape concerns scores were 16.31 with a SD of 6.81 (Table 3).

The BSQ-8C and BES data of the present study did not meet the normality assumption. Hence, we applied Spearman’s correlation rank test, and the results demonstrated a positive correlation (Spearman rho = 0.434, *p* < 0.001) between body shape concerns and binge eating behavior (Table 4).

The body shape concerns among the adolescent girls were significantly higher among the monthly family income category of 5000 to 7000 SAR (ref: less than 5000; Adjusted OR (AOR) = 3.39; 95% CI = 1.45–5.93; *p* = 0.005), those with fathers working in the private sector (ref: government sector job; AOR = 2.69; 95% CI = 1.27–5.74; *p* = 0.010), the severe binge eating categories (ref: none to low; AOR = 4.06; 95% CI = 2.63–6.11; *p* = 0.009), and obese adolescents (ref: normal; AOR = 4.58; 95% CI = 2.11–6.71; *p* = 0.001). The non-significant factors of body shape concerns were age, the education levels of the students, and the education levels of the fathers and mothers (Table 5).

## 4. Discussion

The WHO and the United Nations Children’s Fund (UNICEF) keep adolescent group health as a critical part of the sustainable development agenda and recognize the importance of mental health and well-being [40,41]. The present study was performed in alignment with these organizations’ objectives by assessing critical health problems (binge eating behavior and body shape concerns) among female adolescents in Northern Saudi Arabia, where research in this aspect is limited. 

Regarding binge eating behavior assessed by the BES, we demonstrated that about 78% of the adolescents had no problem controlling their impulses to eat whenever they wanted. Increased impulsivity to eat even without being hungry is directly associated with lower quality of life, negative mood, higher intake of unhealthy food, and other psychological problems, as demonstrated by some authors [42,43,44,45]. Moreover, this could act as a hindering factor in implementing required health promotion interventions for this vulnerable age group [42,46]. Recently, a study by Boswell RG et al. in the United States reported that their study participants with higher impulsivity had higher depressive traits and comorbidities [43]. Our study found that about 18% had varying degrees of problem impulsion. Even though our findings are not as high as those of Boswell RG et al., this sizable proportion of adolescents still faces challenges in these aspects. Similar to Shad HD et al., we found that a considerable proportion (9.5%) of participants had problems with their body weight [47]. The varying degrees of issues across the BES domains indicate that eating behavior is a complex and significant problem among adolescents and necessitates tailored intervention.

Regarding the body shape concerns evaluated by the BSQ-8C, the highest-level problem (one-fourth) explored by the present study was self-awareness of body image in the company of others. These findings can be attributed to the concept of social acceptance, especially during the adolescent period. In this stage, people become more concerned about what others are likely to think of them; thus, they become more conscious, especially in public places or large groups. Therefore, peer pressure and the need for acceptance magnify such concerns, making body image an essential component of adolescence. In contrast, the lowest level of concern was observed in consideration of surgical procedures such as gastrectomies or removal of body fat. This could be due to the self-perception among adolescents of such procedures as being extreme or unnecessary at this age. Our findings underscore the influence of society and friends on the perceived feelings of adolescents’ body image. Our study findings are supported by other studies conducted in different sociocultural settings [48,49,50]. Higher levels of social pressure might lead to negative and low self-esteem behavior among girls. Additionally, a considerable proportion of our study participants followed specific diet and exercise to maintain their body images. In contrast to the present study, Sabiston CM et al. showed different results [51]. Furthermore, their study revealed that body image negative self-consciousness is negatively linked with exercise habits. The variations between the two studies could be due to the differences in sociocultural settings, the tools used, and the participants’ inclusion and exclusion criteria. Even though a small proportion of the girls thought about surgical options to improve their body concerns, this could be due to cultural attitudes, access to healthcare, and self-perception of body image standards. Interestingly, a study by Nerini A et al. among men also reported that sociocultural norms play a significant role in body image concerns and opting for surgical options for social pressure and other reasons [52]. 

Similar to our study’s findings of binge eating behavior, Derks et al. reported that about 12% of participants in their study had subclinical binge eating symptoms [53]. However, in contrast, some authors, including those from the Gulf countries, reported a higher prevalence of binge eating among their study participants, and some studies reported a lower prevalence of binge eating [21,28,54,55]. Additionally, the conservative nature of the Aljouf region might contribute to lower exposure to certain risk factors, such as social media influence, which has been linked to higher binge eating tendencies in other studies. The present study revealed that 8.5%, 7.5%, and 7.8% of the female adolescents in the present study had mild, moderate, and severe body shape concerns, respectively. Furthermore, we found a positive correlation between the BES and BSC-8C scores. The positive correlation between these variables could be due to the vicious cycle involving negative emotions and self-perceptions, which has been documented in earlier works of the literature [27,56,57]. Similar to the variation in binge eating behavior across those studies, body shape concerns and dissatisfaction have also varied in studies conducted in different settings [25,48,58]. Those variations across those studies indicate that binge eating and body shape concerns are influenced by several interconnecting factors, including biological, socioeconomic, and cultural factors, necessitating the evaluation of those factors in different sociocultural settings. It is worth noting that the prevalence of binge eating and body image concerns can vary based upon the tools used for measurement. Moreover, it has been ascertained that binge eating and body image dissatisfaction are no longer problems in Western countries alone; rather, they are global public health concerns [59]. Therefore, tailored educational intervention programs can be planned, as one-size-fits-all is not suitable for this purpose. 

We demonstrated that body shape concerns were significantly associated with family income and the mother’s and father’s working status. Some authors found a positive association with socioeconomic factors. Similar to the present survey, Alharballeh S et al. in the United Arab Emirates [60] and Latiff AA et al. in Malaysia [61] did not find any differences across the age groups of their studies. However, their findings are in contrast with our results regarding the income and occupation of the father. This implies that in Northern Saudi Arabia, other socioeconomic factors, such as income and employment, might influence the perception of adolescents’ body images. In addition, other external social factors could play roles, as the present study did not find a significant association between the father’s and mother’s education and the participants’ body shape concerns. The variations across the studies could be due to differences in the sociocultural settings, the tools used, and the inclusion and exclusion criteria of the participants. Similar to Melisse B et al. [28] and Al-Musharaf S et al. [62], we found that participants with higher problems with binge eating and higher BMIs had significantly higher body image concerns and dissatisfaction in the present study. The notable association between BMI and body shape concerns in the present study could be due to adolescents with high BMIs experiencing body image dissatisfaction due to peer comparison and media projection of unrealistic body images. The roles of media, including social media, in information, education, and health communication activities are well-documented. However, they can mislead the public, especially those who are vulnerable [63,64]. In the current survey of female adolescents, more than one-third of the participants responded that media and social media influence their body image perception. The significant influence of media on body shape concerns among adolescents in our study reflects the crucial roles of media in these conditions. In Saudi society, for instance, where debates on body image can hardly be conducted due to the conservative nature of the society, social media have become the default avenue for information comparison. This might exacerbate the body shape concerns of adolescents, who may want to achieve some of the unrealistic ideals they have obtained on social media. The present study’s findings are consistent with the other studies in this region and Western countries [58,65,66]. This finding reinstates the necessary policy changes that are culturally suitable and adaptable by the local communities.

Our study findings provide a detailed evaluation of binge eating behavior, body shape concerns, and associated factors in a region with limited data on these aspects. The present survey was conducted using the validated BES and BSQ-8C tools to measure binge eating behavior and body shape concerns. This ensured that the data collected were reliable and comparable to other studies. Nonetheless, a few limitations are to be noted by the reader while interpreting the present study findings:Using convenience sampling, we evaluated binge eating behavior and body shape concerns with sociocultural characteristics. This sampling method could have led to selection bias. Hence, the generalization of these findings is likely to be limited.Furthermore, this selection bias might have led to the over-representation of participants from higher socioeconomic backgrounds, particularly in terms of income, education, and occupation, limiting the generalizability of our findings to lower-income populations.The nature of the study design utilized cannot yield a temporal association with body shape concerns and associated factors.We cannot disregard the biases linked with the self-reported survey from this study’s findings.

## 5. Conclusions

The present study observed that a considerable proportion of female adolescents had binge eating behavior and body shape concerns. In binge eating, 20% of the participants had problems with the impulse to eat. The highest level of concern in the BSQ-8C was self-awareness of the body image in the company of others. We found that family income, binge eating problems, the occupations of the parents, obesity, and the influence of social media were significantly associated with body image concerns. The results of the present study can be applied to the development of focused interventions and strategies to address these concerns in this group, with the long-term goal of eventually improving their health outcomes and quality of life. Additionally, it is recommended that a mixed-method survey be conducted among female adolescents in different regions to make region-specific intervention measures.

## Figures and Tables

**Figure 1 nutrients-16-03082-f001:**
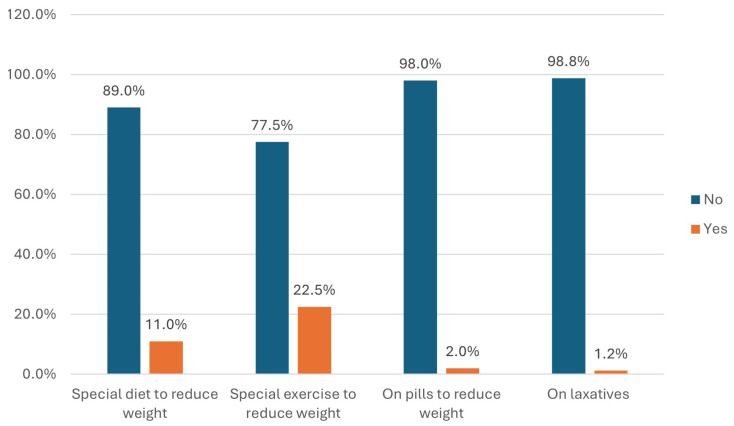
Measures taken by the adolescent girls for their body image maintenance (*n* = 400).

**Table 1 nutrients-16-03082-t001:** Background characteristics of the adolescent girls of this study (*n* = 400).

Variable	Frequency	Proportion
Age (mean ± SD)	15.47 ± 2.83	
Age group, years		
Up to 16	204	51.0
17 and above	196	49.0
Education level (student)		
Up to high school (secondary education) *	225	56.3
Preparatory year **	175	43.7
Occupation status (father)		
Government	224	56.0
Private sector	145	36.3
Unemployed	31	7.7
Occupation status (mother)		
Government	153	38.3
Private sector	203	50.7
Unemployed	44	11.0
Monthly family income (1 USD = 3.75 SAR)		
Less than 5000 SAR	73	18.3
5000 to 7000 SAR	112	28.0
More than 7000 SAR	215	53.7
Education level (father)		
College or more	260	65.0
Up to school	140	35.0
Education level (mother)		
College or more	243	60.7
Up to school	157	39.3
Smoking status		
Nonsmoker	391	97.7
Smoker	9	2.3
Have you been influenced by media (including social media) about body image?		
No	232	58.0
Yes	168	42.0
Obesity status		
Normal	272	68.0
Underweight	33	8.3
Overweight	45	11.2
Obesity	50	12.5

* This includes up to the last stage of school/general education (primary education—first six years, intermediate education—next three years, secondary education—last three years). ** Preparatory (pre-university)—before joining the selected program (higher education at the university).

**Table 2 nutrients-16-03082-t002:** Binge eating behavior categories (*n* = 400).

Variable	Frequency	Proportion
None to low (≤17)	353	88.3
Medium (18 to 26)	22	5.5
High (≥27)	25	6.2
Overall BES score (mean ± SD)	13.27 ± 4.13	

**Table 3 nutrients-16-03082-t003:** Body shape concerns according to the BSQ-8C (*n* = 400).

Variable	Frequency	Proportion
No (<19)	305	76.2
Mild (19 to 25)	34	8.5
Moderate (26 to 33)	30	7.5
Severe (>33)	31	7.8
Overall BSQ-8C scale score (mean ± SD)	16.31 ± 6.81	

**Table 4 nutrients-16-03082-t004:** Spearman’s correlation findings between body shape and binge eating scales.

Variable	rho */*p*-Value
Body shape—binge eating scale	0.434 (<0.001) *

* Significant value (two-tailed test).

**Table 5 nutrients-16-03082-t005:** Predictors of body shape concerns evaluated by binomial logistic regression analysis (*n* = 400).

Variable	Total *n* = 400	Binomial Logistic Regression Analysis			
		No *n* = 305	Yes *n* = 95	Adjusted Odds Ratio (95% CI)	*p*-Value
Age group, years					
<17	204	164	40	Ref	
17 and above	196	141	55	0.69 (0.44–3.34)	0.651
Education level (student)					
Up to high school (secondary education)	225	183	42	Ref	
Preparatory year	175	122	53	2.77 (0.55–6.58)	0.220
Monthly family income (1 USD = 3.75 SAR)					
Less than 5000 SAR	73	59	14	Ref	
5000 to 7000 SAR	112	73	39	3.39 (1.45–5.93)	0.005
More than 7000 SAR	215	173	42	1.93 (0.75–4.93)	0.168
Education level (father)					
College or more	260	203	57	Ref	
Up to school	140	102	38	1.26 (0.62–2.56)	0.517
Education level (mother)					
College or more	243	183	60	Ref	
Up to school	157	122	35	0.49 (0.22–1.09)	0.083
Occupation status of the father					
Government	224	180	44	Ref	
Private sector	145	99	46	2.69 (1.27–5.74)	0.010
Unemployed	31	26	5	0.64 (0.20–2.04)	0.451
Occupation status of the mother					
Government	153	113	40	Ref	
Private sector	203	163	40	0.47 (0.22–0.99)	0.048
Unemployed	44	29	15	1.55 (0.59–4.07)	0.371
Binge eating category					
None to Low	353	283	70	Ref	
Medium	22	11	11	3.33 (1.12–5.95)	0.030
Severe	25	11	14	4.06 (2.63–6.11)	0.009
Obesity status					
Normal	272	226	46	Ref	
Underweight	33	20	13	1.09 (0.62–2.29)	0.092
Overweight	45	37	8	1.73 (0.68–4.39)	0.251
Obese	50	22	28	4.58 (2.11–6.71)	0.001
Media influence					
No	232	200	32	Ref	
Yes	168	105	63	4.24 (2.41–7.46)	0.001

## Data Availability

The raw data supporting the conclusions of this article will be made available by the authors on request.

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
