# Peer review of "Assessment of Binge Eating Behavior, Body Shape Concerns, and Associated Factors among Female Adolescents of Northern Saudi Arabia: A Cross-Sectional Study"

_nutrients, 2024, doi:10.3390/nu16183082_

Round 1

Reviewer 1 Report (Previous Reviewer 1)

Comments and Suggestions for Authors

 I accepted most of your explanations and changes. However, I suggest not including Table no. 6 in the manuscript. There are many reasons for that. First, you, as authors, decide to choose the variables as predictors. I accept your explanation of why body shape concerns were selected as a dependent variable. Second, I think presenting two dependent variables and using each as the independent variable (body shape concerns and binge eating) is a mistake. What is essential is that you have not discussed the results from Table no. 6, and they were not included when the conclusions were written.

I still think that the discussion could be deepening. For example, you wrote (lines 320-321) that body shape concerns were significantly associated with the family income and the father’s working status. What about mothers' and fathers' education? The sentence “No other sociodemographic characteristics were significantly associated with the body shape concerns” is false. The discussion on  SES aspects and body shape concerns would be of great interest.

Line 97. The reference is cited in the wrong way.

Author Response

Reviewer 2 Report (New Reviewer)

Comments and Suggestions for Authors

Overall, I consider this to be a very interesting, well-structured and up-to-date study and I congratulate the authors on their hard work. However, it needs to be slightly improved. In particular, my suggestions are:

- When describing the sample in the method section, the representation of the sample with visual plots with respect to the main attributional variables considered should be improved.

- The results of the reliability of the BES scale (Cronbach's alpha) are not provided in the description of the instrument and should be added.

- When describing the BSQ-8C the authors forget the original creators of the original instrument, the BSQ. They should be cited and commented that the BSQ-8C is a shortened version of the original instrument that maintains adequate psychometric properties.

- The bibliography contains some errors regarding the DOIs of some studies, which are not properly presented.

Author Response

Reviewer 3 Report (New Reviewer)

Comments and Suggestions for Authors

The authors present very interesting data on a highly relevant topic. However, as a physician-scientist working in a Western system, some aspects of the methodology seem unusual to me. Normally, when adolescents participate in a study, informed consent must be obtained not only from a parent or legal guardian (as was done in this study) but also from the participating minor themselves.

Regarding data collection, it appears that both height and weight were self-reported by the participants on the questionnaire. This could introduce bias, as the investigators cannot be certain that the values provided for these parameters are accurate.

The authors mention the use of the 'Binge Eating Scale' (BES), but upon reviewing the references cited to justify the scale's adaptation to the current study population, I found some issues. Reference 28 does not mention the Binge Eating Scale at all. Reference 32 pertains to a study conducted in an obese population in the USA, and the reference itself is quite old (1982). Finally, Reference 33 does indeed validate the Arabic version of the scale, but this was done in a Lebanese population. While this does not necessarily mean the scale cannot be used in the Kingdom of Saudi Arabia, it is difficult to claim that it has been fully validated for this specific population.

Another concern relates to participant recruitment, which was conducted by a data acquisition team in public locations such as shopping complexes and parks. Wouldn’t this introduce selection bias towards individuals with fewer financial concerns? As noted in Table 1, the socio-economic status of the study participants seems fairly high, with more than half of the participants coming from families with an income of over 7,000 SAR. Additionally, the parents appear to have relatively high levels of education. The authors mention this limitation in the discussion section, but it would be prudent to state it more explicitly.

Tables 5 and 6 also need to be checked for formatting issues. In Table 5, the rows describing 'family income' have shifted and do not align with the relevant descriptions in the left column; the same issue exists for 'media influence' in this table. Similarly, in Table 6, the rows describing 'media influence' are also misaligned.

Author Response

This manuscript is a resubmission of an earlier submission. The following is a list of the peer review reports and author responses from that submission.

Round 1

Reviewer 1 Report

Comments and Suggestions for Authors

Dear Authors,

The issue of the relationship between binge eating behavior, body shape concerns, and associated factors among female adolescents is of scientific interest, however, already at this stage, i.e. when specifying the title, it is worth informing what these factors are. After reading the manuscript I have several comments addressed to the authors:

Introduction

In this part, the authors present selected results from previous studies related to the topic of the work, but these results are not presented in a broader context. It is expected that this issue will be shown in a broader context than in the Kingdom of Saudi Arabia (Lines 49-64). In this way, it could be shown that this problem is less recognized in KSA than elsewhere, or that there are other conditions for this phenomenon, which justify undertaking research. Moreover, what is already known in the literature about the relationship between the variables included in the study is not entirely clear. In this way, the Introduction explains the concepts, but there is no justification for why it is worth doing such a study and what the application of the obtained results may be. Why were the "associated factors" not presented in the Introduction, this is also missing from the aim of the study.

Materials and Methods

The description of the methodology is sufficient, although the selection of the study group described in the results should be included in this part (lines 159-160).

Detailed notes:

Why the term “analytical cross-sectional survey” was used? It will be better to use „a cross-sectional survey”

The sentence “The current research included all female adolescents from the Al-Jouf region belonging to the age group defined by the WHO” should be rewritten.

Line 130.  I think that it should be 48 instead of 46

Results

Table 1. For foreigners, it may be difficult to understand the terms used to describe education, e.g. up to school, up to high school, pre-university. It may be worth entering how many years of education there are at each level. I suggest entering income equivalents in brackets, e.g. in dollars. I suggest changing thinness to underweight. 

Table 3. When using the overall BED score, such detailed information is not necessary or can be presented as a supplement. The table does not describe what 0, 1, 2, and 3 mean.

The descriptions of the results presented in Tables 2 and 3 are very selective, which confirms that they are not necessary in the text.

Please explain why the BES score was used as a predictor of body shape concerns and not vice versa.

Maybe it would be worth looking for BES and predictors. Then you could see the differences in the BES and BSQ-8C predictors.

The overall score for BES and BSQ-8C was not presented.

Discussion

The discussion is largely in the form of comparing the results of own study with the results of other studies. There is almost no attempt to explain the results. Many results are cited again, including very detailed ones. However, there are also references to results that were not presented in the results section. For example (lines 229-230) “Even though the majority of the present study participants do not have a problem with this domain (higher impulsivity or depressive traits and comorbidities), about 18% of them had varying degrees of problem impulsion”. In my opinion, the results were not presented for domains, but only about individual questions.

I propose to refine the description of the strengths and weaknesses of the study

Conclusions

I suggest formulating conclusions more precisely. In lines 305 -307, some variables (predictors) are missing.

Reviewer 2 Report

Comments and Suggestions for Authors

1. More than half of the teenage parents in this study, especially the fathers, work in the government sector. Can this sampling result be analyzed?

2. The location and source of subject recruitment were not clearly stated. the author only explained the basic concept for the number of subjects the study recruited. 

3. It's suggested that the author could discuss the characteristiics of the subjects with eating disorder and the differences from other studies.